# Demonstrating the Feasibility of an Economic Empowerment and Health Promotion Intervention among Low-Income Women Affected by HIV in New York City

**DOI:** 10.3390/ijerph20085511

**Published:** 2023-04-14

**Authors:** Prema L. Filippone, Yajaira Hernandez Trejo, Susan S. Witte

**Affiliations:** 1Silver School of Social Work, New York University, New York, NY 10012, USA; 2School of Social Work, Columbia University, New York, NY 10027, USA

**Keywords:** vocational rehabilitation, HIV, employment, prevention, health equity, economic empowerment

## Abstract

Women of color in the U.S. face systematic exclusion from the labor market, work protections, and employer-based benefits. Women’s economic vulnerability increases their susceptibility to health-related issues, including HIV transmission and substance use, which are work-restricting disabilities, by constraining their capacity to effectively reduce risk. The Women’s Economic Empowerment pilot examined the feasibility of a structural intervention, implemented at a neighborhood agency, combining both health promotion and economic empowerment components as a pathway to accessing an urban job market for low-income women with work-restricting disabilities, including living with HIV. Ten women clients from a partner agency in New York completed four health promotion sessions, six financial literacy sessions, and a concurrent opportunity to match savings; some also followed with up to 24 vocational rehabilitation sessions. Interviews captured self-reported data on health promotion and financial outcomes at pre-/post-intervention and 3-month follow-up. Qualitative analysis of recorded group sessions and field notes demonstrate that women express improved HVI/STI knowledge and problem-solving strategies for risk reduction, a shared optimism for the future due to group participation, enhanced social support through relationship-building, a heightened sense of empowerment regarding financial decision making, and a desire to re-engage in the labor force. Findings suggest an empowering approach to re-engage women impacted by poverty, unemployment, and disabilities, including living with HIV, into the workforce may be implemented in a community setting.

## 1. Introduction

People living with HIV/AIDS are living longer and healthier lives yet continue to struggle with significant determinants of health that reduce their opportunities to achieve economic independence, including poverty, unemployment, substance dependence, poor health and care access, disability status, and other adverse events [1]. Due to structural inequities, African-American women, disproportionately represented among people living with HIV/AIDS, are also overly represented among those living with these challenges.

Many women strive for economic independence in their daily lives. Yet, recovery from substance dependence and other adverse events can derail progress towards economic stability, educational attainment, and economic independence. Low-quality jobs and employment instability has been found to be higher among low-income women and a leading factor in financial insecurity [2]. Poor health, work-restricting health conditions, and a markedly diminished need for unskilled labor also contribute to the significant employment gap among communities of color [3,4]. In the United States, one in four Black/African-American adults and one in six Latino/Hispanic adults, respectively, are living with a disability [5]. Among working-age adults (ages 18–64) with disabilities, only 35% had some sort of employment, yet when employed are paid on average 20% less than able-bodied counterparts—low wages being another factor affecting health outcomes [6,7]. New York City adults living with a disability are twice as likely to have incomes below the federal poverty level with the highest disparities among Black and Latino adults and under-employed, working-age women [7,8]. Women living in poverty also experience higher prevalence of mental illness, elevated exposure to violence, and poorer health outcomes, making them more susceptible to several health-related issues, including HIV transmission, by constraining their capacity to effectively reduce risk behaviors [9,10,11]. Social determinants of health such as poverty and unemployment have far-reaching effects on women’s health outcomes.

The next generation of health interventions must consider these individual and structural challenges women face in their daily lives. For example, in examining syndemic, multilevel factors to explain HIV vulnerability among women in the United States, Frew et al. (2017) [12] found financial insecurity to be the most salient feature in underlying sexual concurrency and overall risk-taking. Structural interventions in HIV prevention attempt to shift the HIV risk environment as a means of reducing sexual risk behaviors and improving health outcomes [13,14,15]. Structural interventions that help to economically empower women—through health promotion, financial literacy, and increased work skills—could serve to improve women’s health outcomes and increase their engagement and retention in the labor force, further sustaining economic independence. Combination HIV risk reduction and economic empowerment interventions tested among women at high risk for HIV have demonstrated success in reducing health risk behaviors as well as reducing income from higher-risk behaviors such as sex work [16,17]. The current study examines the feasibility of a structural intervention combining both health promotion and economic empowerment among low-income women with work-restricting disabilities as a pathway to accessing the job market.

A major challenge to structural interventions targeting economic empowerment of low-income women in the United States is barriers imposed by participation in existing welfare or entitlement programs. As women strive for economic stability, many rely on government programs with strict requirements regarding employment status and income. Most government subsidies are time-limited (i.e., Temporary Assistance for Needy Families, Supplemental Nutritional Assistance Program), unless a person has a documented long-term disability. However, even in cases of disability, the income and work regulations are stringent. Federal regulations allow individuals with a disability to work if their earnings do not exceed certain nominal thresholds and hold no more than a few thousand in total assets [18]. If such guidelines are not followed, they may lose future funding and are penalized for every dollar they go over the designated maximum alliance for individual assets. Even if there may be a desire to work and to save money, government regulations may produce a disincentive to both. For women who are recovering from substance dependence and other related adverse life events (such as HIV infection, poor health, intimate partner violence, and mental health concerns), it may be difficult to see the value of engaging in economic empowerment activities with so little economic independence. This paper attempts to explore these issues to enhance our knowledge of effective interventions that support women on disability living in low-income communities and high-risk environments to gain access to the workforce.

The Women’s Economic Empowerment pilot (WEE) described here was initiated to better understand the acceptability and feasibility of implementing a structural intervention to promote women’s economic empowerment in an urban context. Grounded in social cognitive [19,20] and asset theories [21], the intervention targeted behavior change by building women’s self-efficacy in sexual health decision making and condom negotiation and use, while also promoting economic stability through training in financial literacy and the accumulation of economic assets. The study aimed to (1) explore whether a structural intervention combining health promotion with economic empowerment activities would be both acceptable and feasible among women receiving HIV prevention and related case management services in NYC and (2) obtain preliminary data that would support the design of a future efficacy trial. If we better understand the successes and challenges of implementation, we may improve the overall effectiveness of combination, structural economic empowerment interventions for low-income communities. As such, Figure 1 depicts the conceptual model for the combined intervention.

## 2. Methods

The Women’s Economic Empowerment (WEE) pilot was initiated in collaboration with a community-based organization in New York City, providing health and mental health and disability services support to individuals living with low incomes. The pilot study took place between August 2018 and August 2019. Guided by social cognitive and asset theories, the study sought to examine if an intervention combining evidence-based HIV sexual risk reduction sessions with three key economic empowerment components—(1) financial literacy training; (2) vocational training through another local, community-based program; and (3) a matched savings program—would increase women’s HIV/STI risk reduction behaviors and/or engagement in the workforce. The central tenets of social cognitive theory, including self-efficacy and outcome expectancies, have been found to affect whether people consider changing their behavior, the degree of effort they invest in changing, and long-term maintenance of behavior change [19,20]. Self-efficacy with respect to negotiating and using condoms with a partner, for example, has been found to be a strong predictor of condom use and is often found in conjunction with empowerment in sexual relationship decision making. The economic empowerment components for the study have also been adapted to integrate self-efficacy with outcome expectancies related to building financial literacy and vocational knowledge and skills. Asset theory posits that economic assets may yield a range of outcomes, including increased economic stability. These, in turn, may mutually reinforce non-economic assets, including psychological, behavioral, and social assets. For low-income women, or women with disabilities, assets gained from economic empowerment are rich and complex, and have been operationalized to include economic, health, gender-based, and psychological empowerment [21]. The intervention consisted of 4 weekly sessions of HIV Risk Reduction (HIVRR) and 30 sessions of Economic Empowerment. Economic Empowerment comprised 6 weekly sessions of financial literacy, 24 sessions of vocational training provided by a local vocational rehabilitation organization, and a matched savings program. Intervention sessions were conducted for 17 weeks on average, accommodating agency closures resulting from inclement weather and planned holidays.

Recruitment and Enrollment Process. In November 2018, a purposive sample of 17 women invited by agency staff members to be screened for enrollment eligibility was used. Of these, 10 women met the following eligibility criteria: (1) at least 18 years old, (2) proficient in speaking and reading English, (3) has a documented medical disability, learning disability, physical disability, and/or history of substance abuse, (4) has not used drugs other than marijuana in the past 90 days, and (5) was not relocating outside of New York City in the next year. We chose these eligibility criteria to intentionally identify adult, English speaking women who may benefit from an intervention offering behavioral HIV risk reduction and qualifying for free vocational training services due to a medical disability diagnosis. Following screening, eligible women completed an informed consent process. All study procedures and protocol were approved by the study team’s Institutional Review Board.

The study used community-engaged research methods to improve implementation. These included face-to-face planning meetings with agency stakeholders to adapt the interventions; meetings with same to review and discuss assessment measures to best meet the needs of high-risk, low-income women living in NYC; and weekly check-ins with agency administrative staff during study implementation to monitor acceptability of ongoing activities.

Assessments. To demonstrate feasibility to implement assessment measures associated with the pilot, women were asked to complete a 30 min assessment that included questions related to demographics, finances, reproductive health, alcohol and drug use, intimate partner violence, stigma and discrimination, and social support. Self-reported survey data were collected at three different time-points—pre-intervention at screening, post-intervention following completion of all sessions, and a final interview 3 months following the post-intervention assessment. In-person interviews were conducted in English by a trained research assistant. Interviews lasted about 45–60 min in duration and were audio recorded for quality assurance purposes. Question responses were inputted into an electronic survey assessment using a password-protected computer. Study assessment measures asked questions related to demographics, women’s experiences with sexual and drug use risk behaviors, stigma and discrimination, gender-based violence, social support, and financial and employment experiences. Given the small number of participants, for the purposes of this manuscript, only demographic data were summarized in findings. Women received monetary compensation (USD 20–50) following completion of each assessment.

Intervention Sessions. The intervention curricula had been tested in previous research conducted with women at risk for HIV and other STIs in New York City and in Central Asia [15,22]. HIV risk reduction and financial literacy sessions were adapted to accommodate women of color in New York City. A desk adaptation of language and activities was conducted by the team, led by the first author. The adaptations included a focus on locally relevant vocational training and job placement rather than entrepreneurship. While entrepreneurship was consistent within the cultural, business, and economic contexts of Central Asia, it did not reflect the predominant service economy and local culture in which NYC women live. Additional modifications centered on shifts to the language and script that would be more culturally relevant to women of color living in NYC.

Women met weekly at the agency site for 120 min group intervention sessions in a private classroom setting. HIVRR sessions focused on knowledge and skills to reduce HIV/STI risks and provided women with communication and negotiation skills to use with intimate partners. Given the relatively high prevalence of trauma and violence histories reported among women in the study’s target population [23], sessions also aimed to address women’s concerns about intimate partner violence (IPV) related to safe sex practices while underscoring strategies for IPV prevention, including domestic violence and rape. Although the sessions focused on HIV/STI knowledge, overall health behaviors and health-related goal setting were prominent during HIVRR sessions. Table 1 and Table 2 illustrate group component topics covered during HIVRR sessions.

Financial literacy sessions centered on increasing knowledge and skills related to savings; helped women learn about how to use a bank and banking services; educated women about budgets and financial planning strategies; helped women establish a household budget; provided women information and guidelines for debt and loans while reviewing debt management strategies; and helped women develop a long-term savings plan. As part of the financial literacy curriculum, facilitators partnered with a local bank best suited for low-income women providing no-fee banking services and improving potential for savings. A bank representative attended a session, introducing how best to engage services. Women were invited to open a savings account and received matched savings to build incentive for savings behavior as well as accumulate assets to be used towards individual vocational goals. Table 3 and Table 4 depict group component topics addressed through financial literacy training sessions.

HIVRR and financial literacy group sessions were co-facilitated to allow for more individualized attention during group activities, to build in capacity for staff training, implement more individualized skill-building activities, increase opportunities for quality assurance, and reduce staffing gaps. Facilitators collaborated to ensure that different points of view were represented in the discussion and any questions or concerns were addressed and documented for follow-up within the larger group.

Vocational Rehabilitation Services. To ensure access and sustainability, financial literacy sessions were followed by vocational rehabilitation services made available through a community-based program in New York City. Researchers utilized community service providers to link participants to existing community vocational rehabilitation programs that might promote and sustain support for economic empowerment. The vocational rehabilitation program chosen is a state-funded effort designed to assist individuals with disabilities in obtaining sustained employment through vocational guidance, resources, and training. WEE study selection criteria required that all women have a documented medical disability, learning disability, physical disability, and/or history of substance abuse. Therefore, all women interested in vocational rehabilitation services met the agency eligibility criteria.

Agency-trained vocational rehabilitation counselors strictly monitored enrollment and engagement in the vocational rehabilitation program as a natural course of treatment and not as part of the research project. Attendance was verified by the research team using an attendance record signed by counselors and instructors from the vocational rehabilitation program. Women met biweekly with an assigned vocational rehabilitation counselor to establish vocational goals, assess work skills and potential areas of training, and review individual progress. The research team monitored engagement in vocation rehabilitation services through alternating weekly face-to-face meetings and phone-based contact. Women pursued vocational placement training aligned with their employment interests, individual aptitude, and marketable job opportunities such as home healthcare, clerical work, and security services. 

Matched Savings. Matched savings promotes asset building in support of a new vocation or job [16,24]. The WEE project reinforced savings behavior by matching 1:1 any funds earned from attending intervention sessions and subsequently saved in an individual bank account. Each participant was invited to open a personal savings account as part of the financial literacy training sessions. While the research team had no influence over individual bank accounts, they did open and monitor a separate “match savings” account. This account was increased or reduced by an amount comparable to the amount of session incentive that each woman saved in her own personal bank account. For instance, if a woman deposited $20 dollars a week over the course of 4 weeks, at the end of those 4 weeks the research team deposited $80 dollars into a matched savings account. If the next month a woman makes no deposits, but takes out $10 from personal savings, then no additional matched funds were added and instead the matched savings account would be reduced by $10 dollars. To continue to receive the matched savings deposits, each participant was required to attend 4 of 6 financial literacy sessions and 18 of 24 vocational training sessions. Women were free to accumulate personal savings, but if she missed more than the required number of training sessions, she could not to accrue additional matched savings.

Quantitative Analysis. Given the small sample for this feasibility pilot, we share only descriptive summaries related to demographics and services use to characterize the participants’ history and experience with the services system.

Qualitative Analysis. To best capture findings, and assess feasibility and acceptability of the intervention, we rely on qualitative analysis. PF and YH, facilitators, took process notes during sessions and met for debriefing following sessions and with SSW for supervision. Meetings were used to discuss how content was received, which activities seemed most relevant to participants, and any challenges that arose during the group process. Following completion of the sessions, they reviewed digital audio recordings of the sessions and conducted a manual content analysis to identify the most common themes related to the intervention and its impact. Themes were then discussed through an iterative process to identify themes related to intervention feasibility and acceptability. Specific case examples and quotations that illustrated these themes were also identified through this iterative process [25,26,27].

## 3. Results

Demographic and Other Sample Characteristics. The study enrolled a sample of 10 low-income, cis-gender Black and Latina women with a history of drug use, receiving services from a New York City-based community organization. One participant dropped from the study following the first HIVRR session. The mean age of participating women was 50.7 years (range 29–59), and all identified as cisgender and Black or African American. Three women also identified as Latina. Women educationally attained no more than a GED or high school diploma. Although 50% of women had a main partner, most of the group identified as single, never married, and three of the women had at least one child in their household. All women were aware of their HIV status and had been tested for HIV and STIs. Four women had been living with HIV for at least 10 years or more. All women had a history of drug use.

Women’s Financial Status and Household Composition. All women received financial assistance through government programs such as SNAP, TANF, HIV/AIDS services, Disability, and Supplemental Security Income (SSI). As depicted in Table 5, all women received housing support through government-subsidized housing programs as well. Most women received less than $1000 per month and reported no other means of income. Only 3 of 10 women had a bank account, but many also reported savings maintained outside of formal accounts. About 60% of the women did have some form of debt under $500 while only one participant owed $3000 or more. Most debt reported by women was to family members and friends. Other types of debt included outstanding rent or arrears for utilities and store credit debt.

Services Use: As depicted in Table 6, women were highly engaged in community-based services. At baseline, all women had received medical care within the past 90 days and most women (70%) had received a referral for more specialized medical care, such as reproductive health care. High service utilization may in part be because women were recruited from a community-based organization and the entire group had been receiving services for at least 3 months. All the women living with HIV had received services from the CBO. Most (80%) of women had also received HIV counseling or education in the past 90 days. Most women were actively receiving mental health counseling (i.e., psychological counseling and/or domestic violence counseling). Notably, only two women had pursued a job in the last 90 days.

Intervention Attendance. Nine women engaged in all four HIVRR sessions (m = 3.6), eight women engaged in all financial literacy sessions (mean = 5.86), and four women initiated vocational rehabilitation services while three women engaged in an upward range of 18 vocational training sessions (m = 13).

HIV/STI Knowledge Reinforcement. Women had a history of HIV prevention services at the collaborating agency. HIVRR sessions offered women the opportunity to reinforce and improve HIV/STI knowledge. While at baseline women demonstrated good preexisting HIV/STI knowledge, the HIVRR curriculum was adapted to include biomedical innovations such as post-exposure prophylaxis (PEP) and pre-exposure prophylaxis (PrEP). Benefits from HIVRR sessions specifically centered on dispelling myths related to HIV risk levels and STI symptomology. One participant noted, for example, “the colored ones (condoms) do not work as good.” Group discussions were used to sort out misinformation and established evidence-based knowledge and best practices in reducing HIV/STI risks. The most palpable gains in HIV knowledge appeared to be related to STI symptoms and prevention best practices. In session 2 of HIVRR, most women expressed having learned new and useful information from the review of HIV/STI facts. Women expressed marked interest in understanding how to obtain, use, and manage uptake of pre-exposure prophylaxis (PrEP).

Learning the Utility of Problem-Solving Strategies. Women actively discussed sexual risk taking and safe sex negotiation during group sessions. Although half of the sample reported having a main partner, 70% of women reported that they were not sexually active. Two problem-solving strategies were taught: Problem, Option, Plan (POP) and Specific, Measurable, Attainable, Realistic, and Time (SMART). While the intervention was written with the goal of reducing sexual and drug-related HIV transmission risk, women appeared to integrate the problem-solving strategies as a framework for addressing financial decisions and other personal events such as negotiating physical safety with intimate partners and while making educational decisions for a child in school. Safer sex strategies for one woman were used to better negotiate with her sometimes abusive partner and to reduce tension with her partner. She reported: “It’s still a problem but at least I did not get beat up” in using SAFE and POP to avoid violence in her abusive relationship. She noted that through these tools she further realized that initiating the sexual encounter allowed her to reduce her risk.

“In order for you to come to the class you know when you get home you either have to give him sexual pleasure, for him, to not, to avoid getting physical and angry.”

Another woman described how she used POP with her daughter’s guidance counselor as they worked to solve a problem in the classroom. Some of the women used the risk-reduction strategies (such as avoidance, self-talk, and confronting negative thoughts) to circumvent anticipated triggers for drug use relapse. One woman mentioned,

“2003 came from Long Island, couldn’t go back to Long Island because I knew if I went back people, places and things, so that’s why I had to search and fine me an apartment. I’ve been 16 years clean.”

Women also discussed using self-talk when they see instances of people on the street “nodding” which reminded them of their own physical behavior when they were high on drugs. Women discussed the importance of using SAFE, POP, and the trigger tools to avoid situations and drugs that are often precipitate sexual risk-taking behaviors. Women also used problem solving throughout the financial literacy sessions, such as when someone discussed arguing against negative thoughts in response to how they could be sure that they would stay on a monthly budget. Additionally, the POP tool was used for daily decision-making needs that the women had, such as when one participant was trying to ensure she stayed on budget for her daughter’s birthday party.

Potential to Reengage in the Labor Force. Despite having been unemployed after temporary or long-term disability, nearly half of the women in the study (n = 4) participated in individualized vocational rehabilitation services with the goal of reengagement in the labor force. Women who pursued vocational training ranged from 28–50 years old. Women completed an average of 17.5 sessions of vocational training, and sessions consisted of recertification classes for para-professional licensure, broad individualized job placement training sessions, and the pursuit of education-based goals to expand employment opportunities. Below, we share more detail regarding the experience of two women in the program and how they utilized the session materials to achieve their goals.

### 3.1. Case Study 1

Janessa is a 28-year-old woman receiving TANF benefits due to housing insecurity. She experienced unstable housing, unemployment, and a violent intimate partner relationship. She was engaged in all components of the intervention, participating in a total of 4 HIVRR sessions, 6 financial literacy sessions, and 17 vocational training sessions. During the sessions, she was eager to expand her HIV/STI knowledge and financial literacy knowledge and build related risk-reduction skills. Over the course of the intervention, she described integrating her skills-based learning into daily activities. During HIVRR sessions, she shared that learning HIV/STI knowledge dispelled misconceptions of HIV/STI risk and motivated her to prioritize learning strategies to make healthier decisions. She also integrated problem-solving and goal-setting strategies. She was negotiating housing insecurity, as her landlord was selling her building, and economic instability due to increasing financial demands. She used the financial literacy sessions to learn and engage in budgeting and then established a savings plan to cover her moving expenses. During sessions, facilitators checked in with each woman to tailor activities and information supporting her goals. Access to personalized banking services empowered Janessa to open a bank account after years of concern about stigmatization and exclusionary practices. Janessa established regular contact with members as support in everyday decisions. She voiced a desire to come to group despite some difficult personal days. Facilitators noticed that as Janessa set and reviewed her goals, she was able to build agency in making ongoing changes for herself through the support and reinforcement of her peers. Additionally, a more individualized approach to the facilitation process provided reinforcement of content, but also allowed facilitators to be responsive to women’s needs, to adapt our approach to goal setting, and to establish more personalized monitoring of progress towards individual goals.

### 3.2. Case Study 2

Roselyn is 50 years old and receives social security benefits because of a long-term physical disability and is HIV positive. She was an enthusiastic and crucial participant in the pilot, attending 4 HIVRR sessions, 4 financial literacy sessions, and 18 vocational training sessions. Roselyn contributed to every session she attended and specifically demonstrated a keen interest in learning new knowledge of HIV/STI and its developments, financial literacy, and vocational training. Specifically, related to HIVRR, she shared the importance of gaining new knowledge such as the use of PrEP as a prevention tool and gaining clarity on myths and facts of HIV/STI. Roselyn participated in POP as a tool to assist in problem-solving issues as they surfaced in her intimate relationship. During the sessions, she had started dating someone in her neighborhood, but felt as though revealing her HIV status would significantly impact intimacy so she focused on abstinence first. Over time, she expressed an interested in gaining knowledge about safer sex practices. The HIV risk reduction curriculum provided her with clear alternatives for safer sex practices. She used POP to incorporate more intimacy in her relationship by brainstorming scenarios and establishing best practices learned during the sessions. Financial literacy sessions helped her establish budgeting goals as she sought to save for the future and purchase various items for her home. Sessions also provided Roselyn guidance on developing a savings plan to ensure her immediate needs were addressed as she regularly added to savings. Initially, Roselyn was unsure of engaging in VR as she perceived her age and disability as a barrier to employment. As sessions progressed, she discovered different interests and the desire to engage in vocational rehabilitation to uncover potential skills and abilities. The pilot encouraged Roselyn to continue exploring job placement opportunities, and by the end of the intervention, she was able to complete 18 vocational sessions including job placement evaluation sessions, developed an individualized plan, and established employment objectives. Roselyn ultimately enrolled in the Trial Work Period as part of the SSDI work incentive. She has expressed an optimistic outlook and shared excitement at the opportunity to work despite her disability.

Enhanced social support and improved outlook. These two case studies illustrate the ways in which the intervention helped to improve women’s expectancies and self-efficacy and facilitated opportunities to start thinking about the future. For some, this was the first time in many years they reconsidered employment. Roselyn noted that this was the first time she thought about work in almost twenty years. In one session, she shared:

“I was like depressed. All I was doing sleeping feeling bad all day. My asthma was real bad. Ain’t nothin’ to do or get me outside. This group got me out, ya. Hadn’t thought about no work no job…didn’t think I could… not in a long time. Not since I was like 30. Felt real good to have something to do.”

A prolonged dislocation from the workforce combined with adverse life events and chronic health issues deterred Roselyn from thinking about work as a viable option despite the work incentive program. Intervention sessions served as a catalyst to help women reconnect with the idea of work and to think about future plans by creating structure in their lives and also additional encouragement and reinforcement for change.

## 4. Discussion

Social determinants of health such as intergenerational poverty, lack of educational opportunities, unemployment, and disparate access to quality healthcare serve as key drivers in the reproduction of health disparities and income inequality for communities of color. Structural-level activities combining health promotion with economic empowerment may strengthen interventions to succeed; as we consider structural interventions to improve health outcomes, it is important to consider barriers.

The findings of this pilot study demonstrate that when combination interventions offer both health risk reduction and economic empowerment, women may achieve both improved health and economic outcomes. Reentry into the work force in a meaningful and sustained way may extend gains by ensuring ongoing economic support, which may further help reduce health risk behaviors. The social support and encouragement women received by the group were crucial to facilitating engagement in the intervention. Although only a third of the sample participated in the vocational training sessions, the women who engaged in all aspects of the intervention expressed a shared optimism for the future and an active reengagement with the labor force.

We learned several lessons during this pilot study regarding collaboration with community partners to achieve implementation success. Collaboration with an agency was essential as it expedited participant buy-in to the pilot, allowed for more flexibility and responsiveness to women’s needs, and further improved feasibility for future implementation. Pre-COVID, our community partner expressed interest in continuing to offer the program. Utilizing an existing vocational training services component offered an established, well-known service, which speaks to longer-term sustainability of such an intervention. Programming was very individualized, in which women’s specific needs and concerns related to their disability could be addressed, but it also meant that the duration of the vocational training component had to be adjusted as there were periods in which women were waiting to begin programming or awaiting verification of disability documentation and medical clearance forms, etc., which required a level of monitoring that paralleled social support services with regular check ins over the phone and in person.

Limitations. The small, purposive sample selected for this pilot study necessarily limits any conclusions to be drawn beyond the feasibility and acceptability of the work by this sample of women receiving services at a neighborhood agency in Queens, New York.

## 5. Conclusions

The Women’s Economic Empowerment pilot demonstrated the feasibility and acceptability of a structural intervention, implemented at a neighborhood agency, combining both health promotion and economic empowerment components as a pathway to accessing an urban job market for low-income women with work-restricting disabilities, including living with HIV. Given the pervasive and systematic exclusion of women of color from the labor market, from work protections, and employer-based benefits in the U.S., and their susceptibility to health-related issues, including HIV transmission and substance use, such interventions could play an important role in increasing their access to employment. Findings suggest that such an empowering approach to re-engage women impacted by poverty, unemployment, and disabilities, including living with HIV, into the workforce may be implemented in a community setting. Future work to test efficacy of this intervention should continue to center community-engaged processes and should expand such opportunities to non-English-speaking participants and other groups in need of economic empowerment, including migrant or immigrant individuals.

## Figures and Tables

**Figure 1 ijerph-20-05511-f001:**
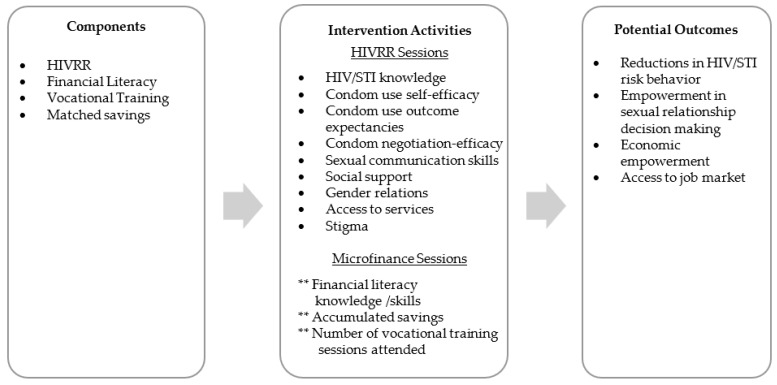
Conceptual model illustrating combination HIVRR and microfinance intervention. ** Uniquely associated with asset theory.

**Table 1 ijerph-20-05511-t001:** Health promotion components integrated into HIVRR sessions.

Health Promotion Group Components	Session # *			
Knowledge/Awareness-Raising	One	Two	Three	Four
Facts about HIV, HCV and other STIs–Review common myths about HIV, HCV and STIs, screening of a video on common STIs and transmission risks, distribute a fact sheet which summarizes information on STIs, review information in jeopardy-style group quiz game		X		X
Differences in partners–Identify and acknowledge the different relationships participants have with different sexual partners	X			
Identify personal risks for HIV, HCV and other STIs	X			
Alternatives to unsafe sex–Show participants options to unprotected sex that can be fun and engaging using Café Menu list.		X		X
Barriers to condom use, getting comfortable with condoms		X		X
Barriers to discussing sex with partners		X		
Risk in long-term relationships		X		X
Drug & alcohol use during sex-Discussion about triggers and personal risks			X	X
Triggers for unsafe sex–Discuss definition of triggers, written activity for participants to identify their own triggers			X	
Intimate Partner Violence (IPV)/Gender-based violence (GBV) & health consequences			X	
Behavioral Skills				
Using condoms–Review steps to using male condom & female condoms practice with models.		X		
Reflective listening technique–Learn steps of this communication technique		X		X
Turn around refusal–Learn to refuse unsafe sex and negotiate condom use in a way that does not anger intimate or paying partners			X	X
Negotiating safer sex in challenging situations when feeling unsafe or at risk of IPV/GBV			X	X

* HIVRR sessions were conducted sequentially from one to four. # refers to the session number in which each health promotion component occurred during the HIVRR sessions.

**Table 2 ijerph-20-05511-t002:** Health promotion components integrated into HIVRR sessions (continued).

Health Promotion Group Components	Session # *			
Self-Efficacy/Empowerment	One	Two	Three	Four
Goal-Setting–Provide opportunity to practice new skills at home	X	X	X	X
Identifying positive reasons to stay healthy through safer choices	X			
Health Quiz-Reinforce health knowledge gained during HIVRR sessions in a fun engaging exercise				X
Harm reduction philosophy-Define a harm reduction philosophy to support steps toward positive change				X
Safety Planning–Learning to recognize consequences of abusive behavior by any sexual partner, police or others			X	
Linkage to Care & Social Support				
Identification of Social Networks	X			
Identification of HIV/STI transmission risks within social network	X			
Identification of institutions that provide formal support within the community	X		X	X
Safety Planning–Identify participants who may be at risk for IPV or GBV and to link them to appropriate services	X	X	X	X
Participant Engagement/Housekeeping				
Welcome/opening/icebreaker-Engage participants to attend the intervention sessions	X	X	X	X
Purpose of Project-Review expectations of their participation in project and why intervention needed	X			
Pros/cons of participating-Familiarize participants with the benefits of this project and possible challenges	X			
Ground rules-Familiarize participants with the procedures and guidelines of the project and develop social contact	X			
Closing ceremony-Reward and celebrate completion of the program				X
Evaluation-Obtain participants feedback on HIVRR component of intervention				X

* HIVRR sessions were conducted sequentially from one to four. # refers to the session number in which each health promotion component occurred during the HIVRR sessions.

**Table 3 ijerph-20-05511-t003:** Group components integrated into financial training sessions.

Group Components	Session # *					
Knowledge/Awareness Raising	One	Two	Three	Four	Five	Six
Facts about savings- Learn what are savings, discuss perceptions and benefits of savings	X	X	X			
Financial institutions–Identify myths, characteristics, and purposes of banks and other financial institutions		X	X			
Budgeting–Define budget, identify household expenses, monetary income and output, and steps to achieve goals		X	X	X		
Budget adherence–Define and identify difficulties to stay on a budget-identify ways to cut spending, prioritize needs and adhere to budgeting plan				X	X	
Capital resources–Differentiate participants’ own capital from loan capital and different sources of income					X	
Loans–Identify trusted financial institutions for loans, make informed decisions about taking a loan, and learn vocabulary and concepts discuss with lenders					X	
Debt management–Identify reasons to borrow money, build steps to pay financial debts, and identify manageable debt					X	
Save for emergencies-Identify types of emergencies and calculate funds needed to respond to these emergencies	X					X
Skills Building						
Bank/Saving account–Practice using the formal banking system by opening a savings account at a local/partner bank	X	X				
Saving money–Practice regularly setting aside money to deposit into savings account	X	X	X	X	X	X
Organizational skills—weekly planning worksheets, prioritize expenses, and execute financial goals	X	X	X	X	X	X
Financial decision making–Learning to prioritize, spend within a budget, and assess needs for loan-taking	X		X			
Navigating financial institutions-Gain comfort in approaching a bank, a lender, or other financial institution and inquire about services and follow through	X			X	X	

* Financial training sessions were conducted sequentially from one to six. # refers to the session number in which each group component occurred during the financial training sessions.

**Table 4 ijerph-20-05511-t004:** Group components integrated into financial training sessions (continued).

Group Components	Session # *					
Skills Building (Continued)	One	Two	Three	Four	Five	Six
Budgeting–Develop skills to stay on budget, cut expenses, buy less on credit, and get family on board			X	X	X	
Emergency planning–Prepare for financial emergencies in the family						X
Self-efficacy/Empowerment						
Goal-Setting-Reinforce goals and financial plans tailored to individual needs and acquire the steps to achieve financial goals	X	X	X	X	X	X
Match funds–Cash available allows participants to use money to buy tools for their vocational training	X	X	X	X	X	X
Assertiveness–Gain vocabulary to discuss financial matters with banks lenders, and other financial institutions			X	X	X	
Linkage to Care & Social Support						
Identification of social support–Help participants identify trusted friends and families that provide support savings plan			X	X		
Safety Planning–Identify and plan for partner’s or family’s threatening behaviors in the face of participants’ new asset building skills	X	X	X	X	X	X
Participant Engagement/Housekeeping						
Welcome/opening/icebreaker/session preview–Engage and increase motivation to attending sessions	X	X	X	X		
Ground rules–Familiarize participants with the procedures/guidelines of the project and develop social contract	X					
Matched savings–Identify benefits and logistics of match savings	X					
Closing–Reward & congratulate participants for program completion						X
Evaluation–Obtain participants feedback on financial literacy component of the microfinance intervention						X

* Financial training sessions were conducted sequentially from one to six. # refers to the session number in which each group component occurred during the financial training sessions.

**Table 5 ijerph-20-05511-t005:** Financial and household composition (N = 10).

Characteristic	n	%
**Total Monthly Income**		
$1000 or less	10	100
**Specified Source of Income**		
Stipend	10	100
Disability	2	20
HIV/AIDS Services (HASA)	2	20
Supplemental Security Income	2	20
Supplemental Nutritional Assistance Program (SNAP)	8	80
Temporary Assistance for Needy Families (TANF)		
Other	7	70
**Housing**		
Apartment rental (subsidized housing)	10	100
**Own a bank account**	3	30
**Savings**		
Yes	3	30
No	7	70
**Location of savings**		
Bank account	2	20
Other	1	10
**Debt**	6	60
No debt	4	40
Less than $500	5	50
$3000 or more	1	10
**Type of Debt**		
Family members	2	20
Store credit	1	10
Friends	2	20
Other–Utilities and rent	3	30
**Women with at least one dependent**	3	30

**Table 6 ijerph-20-05511-t006:** Service utilization over the past 3 months (N = 10).

Types of Services	n	%
HIV counseling or education	8	80
HCV counseling or education	2	20
Counseling or education for other STIs	7	70
Legal help	4	40
Domestic violence counseling	3	30
Psychological counseling	6	60
Education or job training	1	10
Housing assistance	3	30
Help obtaining public benefits	7	70
Help finding a job	2	20
Birth control counseling or education	1	10
Gynecological or reproductive health care	6	60
Primary medical care	10	100
Referral to any other specialized medical care that you may need	7	70
Childcare services	0	0
Medical care for your children	1	10
Drug treatment	0	0
Hx of Drug Treatment	3	30

## Data Availability

Data is available upon request from the corresponding author.

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
