# Peer review of "Demonstrating the Feasibility of an Economic Empowerment and Health Promotion Intervention among Low-Income Women Affected by HIV in New York City"

_ijerph, 2023, doi:10.3390/ijerph20085511_

Round 1

Reviewer 1 Report

I was fortunate to see this presentation at APHA several years ago. This is an interesting project and has great potential to bring benefit. The details of the components of the intervention are not clear, and it is not clear how the feasibility and acceptability were evaluated. Tables may help condense the information. I do think this project is valuable and look forward to learning about the efficacy of the intervention later. The project itself is worthwhile and I'm excited to see the data published. However, this manuscript is overly loaded in the introduction and it is unclear what the components of the intervention were (in detail) and how feasibility and acceptability were assessed. If this is a case where there is too much data to include the pertinent details in a single manuscript, the authors may need to generate separate manuscripts for the different components, and then one connecting everything together. 

Author Response

Dear Reviewer 1,

We appreciate your thoughtful comments and attend to your critique below as follows:

The details of the components of the intervention are not clear, and it is not clear how the feasibility and acceptability were evaluated. Tables may help condense the information. 

We have added tables 1-4 that describe and define the intervention activities and hope that this makes the components clearer.

However, this manuscript is overly loaded in the introduction and it is unclear what the components of the intervention were (in detail) and how feasibility and acceptability were assessed. 

We have edited the introduction and hope that it reads more clearly. 

We have also clarified that we used analysis of the transcripts of qualitative data – from digital recordings of women’s participation in the sessions and facilitator notes and observations - to assess feasibility and acceptability. 

Reviewer 2 Report

This paper reports a mixed-method pre- and post evaluation outputs, outcomes and short-term impact of the pilot program-Women's Economic Empowerment (Aug 2018-Aug 2019) targeting women in NY with a documented medical/learning/physical disability or a history of substance use but no recent drug use in the previous 3 months except marijuana. The structural intervention tackles intersectional disadvantages of financial, employment and health literacy and wellbeing. While primarily aiming at promoting women's self-efficacy in sexual health decision making and condom negotiation and use skills, the pilot intervention also incorporates financial literacy and capabilities to accumulate financial assets. 

I recommend this type of intersectional structural interventions among disadvantaged women who are potentially at risk of HIV and other sexually transmissible infections.

For some minor improvements (I will let the authors to consider):

The main point would be to clearly state why the eligibility criteria (across disability, problematic drug use etc.) were set in correspondence to the designated intervention outcomes and impact? 

A further point would be was there/will there be any co-design with women in any stage in light of empowering women who are heavy social welfare service users but also who are quite disadvanaged financially and health-wise?

The final minor point would be consider non-English speaking migrant women in future expansion/scale-up?

Author Response

Dear Reviewer 2,

We appreciate your thoughtful comments and attend to your critique below as follows:

The main point would be to clearly state why the eligibility criteria (across disability, problematic drug use etc.) were set in correspondence to the designated intervention outcomes and impact? 

We clarified why we determined the eligibility criteria for the study.

A further point would be was there/will there be any co-design with women in any stage in light of empowering women who are heavy social welfare service users but also who are quite disadvantaged financially and health-wise?

We have added the suggestion to incorporate more feedback from participants and in the future to propose additional community collaborative or community-engaged processes to adapt, refine, and then test the intervention formally.

The final minor point would be consider non-English speaking migrant women in future expansion/scale-up?

We appreciate this recommendation for future work and added this sentence to the Conclusions section.

Reviewer 3 Report

Please briefly describe what is the social cognitive & asset theories.

Who is the partner agency?

How did the authors asses the  following eligibility criteria: 1. whether or not a medical disability, learning, physical, history of substance use was met; 2. has not used drugs other than marijuana the past 90 days.

The authors can present the demographics descriptively of the women who participated in the study but the sample size is not enough to justify any tables. At best, the findings can be described qualitatively but NOT quantitatively

What was the response rate for the study? Why were only 10 women enrolled? What was the preliminary size of the population that the authors were trying to reach & ultimately asses?

The authors fail to report any limitations with the study.

Author Response

Dear Reviewer 3,

We appreciate your thoughtful comments and attend to your critique below as follows:

Please briefly describe what is the social cognitive & asset theories.

We have added a brief description of social cognitive and asset theories.

Who is the partner agency?

We choose not to name the partner agency in a published piece, but we can certainly share the name of the agency with the reviewer. 

How did the authors assess the  following eligibility criteria: 1. whether or not a medical disability, learning, physical, history of substance use was met; 2. has not used drugs other than marijuana the past 90 days.

As noted in the document, screening was based on self-report.

The authors can present the demographics descriptively of the women who participated in the study but the sample size is not enough to justify any tables. At best, the findings can be described qualitatively but NOT quantitatively.

We have removed the demographics table and now only describe the sample of women qualitatively.

 What was the response rate for the study? Why were only 10 women enrolled? What was the preliminary size of the population that the authors were trying to reach & ultimately assess?

As this was a pilot study, we selected a purposive sample of 17 women for this feasibility study. While we had originally hoped that all 17 women would attend the intervention sessions, only 10 were able to attend. In a future intervention trial, we will identify a large enough sample for generalizability to the population.

The authors fail to report any limitations with the study.

We have added a limitations paragraph.

Round 2

Reviewer 1 Report

The addition of the underlying theory and tables describing the content help dramatically. One more addition I would appreciate is a figure portraying how the behavioral empowerment and economic empowerment skills work together and how the skills taught can be used in each realm. 

Author Response

Please note the manuscript has been updated to now include Figure 1 attached with the intervention conceptual model and an additional sentence [line 106].

Reviewer 3 Report

The authors have addressed my comments

Author Response

Thank you again for your time and consideration of our revisions. Per Reviewer 3, there are no outstanding issue to address related to this second round of revisions. 

Please see attached manuscript that has been updated to include a conceptual model.
